# Epigenetic models developed for plains zebras predict age in domestic horses and endangered equids

Brenda Larison [1,2,13✉], Gabriela M. Pinho [1,13], Amin Haghani[3], Joseph A. Zoller [3], Caesar Z. Li [3], Carrie J. Finno[4], Colin Farrell[5], Christopher B. Kaelin [6,7], Gregory S. Barsh [6,7], Bernard Wooding[8], Todd R. Robeck [9], Dewey Maddox[10], Matteo Pellegrini[5] & Steve Horvath [3,11,12✉]

Effective conservation and management of threatened wildlife populations require an accurate assessment of age structure to estimate demographic trends and population viability. Epigenetic aging models are promising developments because they estimate individual age with high accuracy, accurately predict age in related species, and do not require invasive sampling or intensive long-term studies. Using blood and biopsy samples from known age plains zebras (*Equus quagga*), we model epigenetic aging using two approaches: the epigenetic clock (EC) and the epigenetic pacemaker (EPM). The plains zebra EC has the potential for broad application within the genus *Equus* given that five of the seven extant wild species of the genus are threatened. We test the EC's ability to predict age in sister taxa, including two endangered species and the more distantly related domestic horse, demonstrating high accuracy in all cases. By comparing chronological and estimated age in plains zebras, we investigate age acceleration as a proxy of health status. An interaction between chronological age and inbreeding is associated with age acceleration estimated by the EPM, suggesting a cumulative effect of inbreeding on biological aging throughout life.

[1] Department of Ecology and Evolutionary Biology, University of California, Los Angeles, CA 90095, USA. [2] Center for Tropical Research, Institute of the Environment and Sustainability, University of California, Los Angeles, CA 90095, USA. [3] Human Genetics, David Geffen School of Medicine, University of California, Los Angeles, CA 90095, USA. [4] Department of Population Health and Reproduction, School of Veterinary Medicine, University of California, Davis, CA 95616, USA. [5] Department of Molecular, Cell and Developmental Biology, University of California, Los Angeles, CA, USA. [6] HudsonAlpha Institute for Biotechnology, Huntsville, AL 35806, USA. [7] Department of Genetics, Stanford University, Stanford, CA 94305, USA. [8] Quagga Project, Elandsberg Farms, Hermon 7308, South Africa. [9] Zoological Operations, SeaWorld Parks and Entertainment, 7007 SeaWorld Drive, Orlando, FL, USA. [10] White Oak Conservation, 581705 White Oak Road, Yulee, FL 32097, USA. [11] Department of Biostatistics, Fielding School of Public Health, University of California, Los Angeles, Los Angeles, CA, USA. [12] Altos Labs, San Diego, CA, USA. [13] These authors contributed equally: Brenda Larison and Gabriela M. Pinho. ✉email: blarison@ucla.edu; shorvath@mednet.ucla.edu

Effective management of threatened species relies on the ability to estimate demographic trends, which depend, in turn, on accurate information about age distributions within populations[1]. Growth rates shape age distributions, can reflect past and current environmental and anthropogenic perturbances[2,3], and can also be used to predict future population growth[4]. However, age is challenging to quantify in wild animals. Age estimation typically requires either investment in long-term field studies or invasive approaches that may not be feasible in live animals[4,5]. Another problem is the limited accuracy of some methods, which may negatively impact conservation efforts[2]. The challenges and importance of obtaining accurate age information have motivated efforts to develop an accurate and non-invasive approach to aging wild animals[4,6].

Epigenetic aging models, particularly epigenetic clocks (ECs), promise to improve the aging of wild animals and thereby make valuable contributions to wildlife conservation and population biology[4,6]. These highly accurate clocks use information from genomic methylation patterns and have been studied extensively in humans[7–9] and mice[10–12]. The development and use of epigenetic models in other species are still limited but are becoming increasingly common[6]. A critical limitation to developing epigenetic aging models for wildlife is that these models need to be trained on samples from individuals of known age. Therefore populations of non-model organisms with known-age individuals are of extreme importance[13]. Here, we develop epigenetic models for a wild equid (plains zebras, *Equus quagga*) using both blood and remote biopsy samples collected from known-age individuals in a captive-bred population.

Besides their high accuracy, four other features of epigenetic aging models should make them attractive for wildlife managers. First, they can be developed from many different tissue types[6,14]. Second, they can be created based on very few genomic sites. The most accurate clocks for humans involve only a few hundred CpG sites[14,15], and far fewer CpGs have been used to build ECs in some wild vertebrates[3,16,17]. Third, an accurate clock can be developed using relatively few individuals of known age[3,16,17]. Finally, ECs developed for one species have been shown to predict age accurately in closely related taxa (e.g., humans and chimps[14]), and therefore can be developed for a less threatened species with the intent of using them in threatened sister species. Using this rationale, we aim to test the performance of the EC developed for plains zebras to predict age in domestic horses (*E. caballus*) and two threatened species: Grevy's zebras (*E. grevyi*) and Somali asses (*E. africanus somaliensis*).

Since individual chronological ages are known for our plains zebra population, it is possible to estimate how fast individuals are aging compared to others by the discrepancy between epigenetic age and chronological age, dubbed age acceleration. Positive age acceleration indicates that an individual is biologically older than expected based on its chronological age. Age acceleration is predictive of all-cause mortality in humans[18–22], which suggests that epigenetic models can be a powerful approach to study the impact of different factors on biological aging. Age-acceleration has also been associated with stress and adversity[23–25], elevated glucocorticoids[26,27], and inbreeding[28–31], all of which are relevant for managing wild populations. The plains zebra population sampled in this study has a complex pedigree due to semi-captive breeding, which creates the opportunity to test whether inbreeding is associated with accelerated epigenetic aging in this population. The strong correlation of ECs to age can sometimes negatively affect their ability to detect age acceleration associated with biological variation[32,33]. We therefore also develop a second model for the plains zebra, the epigenetic pacemaker model (EPM). The EPM has previously been found to be useful for investigating how environmental and life-history factors influence aging[34,35].

Our main goals for this study are to (1) develop epigenetic aging models for plains zebras; (2) test the ability of an EC developed for plains zebras to predict age in three equid species; and (3) estimate the influence of inbreeding levels on plains zebra aging patterns by correlating it with age acceleration predicted by both an EC and EPM. Our main results include an EC that predicts age accurately in plains zebras and three congeners tested, including domestic horse and two endangered sister species. We further show that inbreeding associated age acceleration increases with age, suggesting that inbreeding may have a cumulative effect on age acceleration throughout life. The development of epigenetic aging models in a wild equid stands to have broad impact because the crown group of the genus *Equus* comprises a closely related group of six extant species, five of which range from near threatened to critically endangered[36,37].

## Results

**Epigenetic aging models**. We developed epigenetic models using methylation data profiles from three plains zebra data sets: (1) 76 blood samples, (2) 20 biopsy samples, and (3) 96 blood and biopsy samples combined (Table 1). For each data set, we developed both an EC and an EPM. We evaluated the effectiveness of applying the blood-based zebra EC to predict age in other equids using known-age domestic horses (*E. caballus*, $n = 188$), Grevy's zebras (*E. grevyi*, $n = 5$), and Somali asses (*E. africanus somaliensis*, $n = 7$).

To develop the ECs we fit a generalized linear model with elastic-net penalization using leave-one-out (LOO) cross-validation. To improve EC fit[14] we square root transformed chronological age prior to fitting the models. The blood EC (Pearson's $r = 0.96$, median absolute error (MAE) = 0.56 years, Fig. 1a) and the combined tissue EC ($r = 0.89$, MAE = 0.62, Fig. 1c) predicted age more accurately than the biopsy EC ($r = 0.62$, MAE = 1.79 years, Fig. 1b). The blood EC selected 70 CpG sites, the biopsy clock 31 CpGs, and the combined clock selected 99 CpGs. The biopsy EC had no CpG sites in common with the blood and combined ECs. The blood EC and combined tissue EC shared only two CpGs. We report coefficients, intercepts, and lambdas in Supplementary Data 1 and present the results from using untransformed ages in Supplementary Fig. 1.

Cross-species predictive ability was high (Fig. 1d–f). The zebra-blood EC predicted horse age with high accuracy ($r = 0.93$, MAE = 1.82). The error when predicting the ages of horses younger than 15 years is lower (MAE 1.15) than when predicting the ages of older horses (MAE = 3.97). While prediction errors for Grevy's zebra and Somali wild ass ages were even lower (MAE of 1.08 and 1.15 respectively), this should be viewed with caution as we had limited sample sizes from Grevy's zebra ($n = 5$) and Somali wild ass ($n = 7$).

To construct plains zebra EPMs we used sites in which methylation levels were highly correlated with individual chronological age. Epigenetic states were estimated using a leave-one-out cross-validation. The blood, biopsy, and combined-tissue EPMs included 391, 242, and 248 CpG sites, respectively. We provide details of the CpGs selected by each model in

### Table 1 Description of the zebra data.

| Tissue | N | No. female | Mean age | Min. age | Max. age |
|--------|----|-----------|----------|----------|----------|
| Blood | 76 | 42 | 5.21 | 0.156 | 20.2 |
| Biopsy | 20 | 9 | 5.87 | 0.162 | 24.8 |

We restrict the description to animals whose ages could be estimated with high confidence (90% or higher). Tissue type, N = Total number of samples/arrays. Number of females. Age: mean, minimum and maximum.

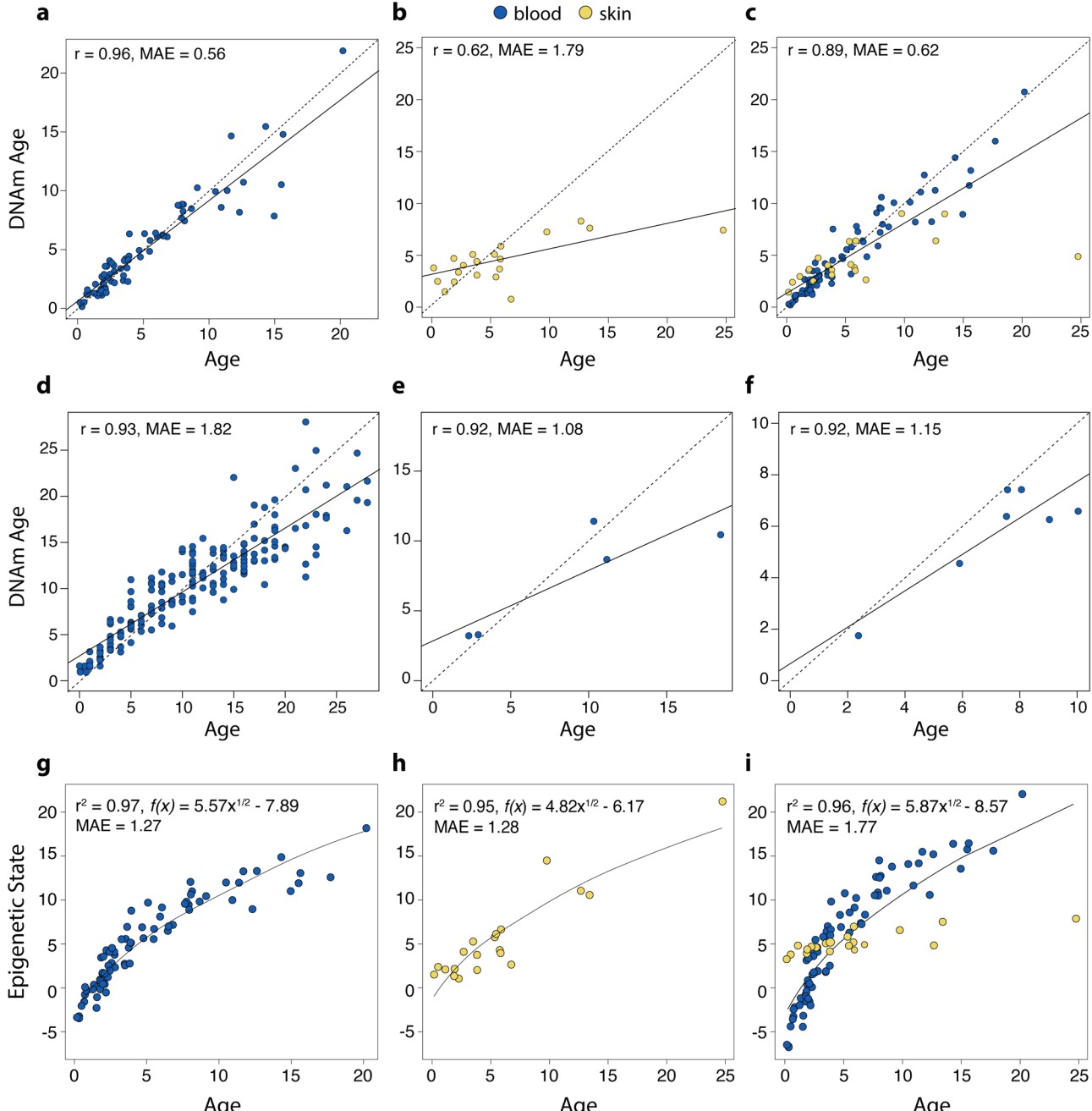

**Fig. 1 Predictive ability of epigenetic aging models. a–c** Plains zebra epigenetic clocks (EC). We developed 3 ECs for plains zebras using square-root transformed ages: **a** blood samples ($n = 76$), **b** biopsy samples ($n = 20$), and **c** combined tissue types. Leave-one-sample-out (LOO) estimate of DNA methylation age are plotted against chronological age. Linear regressions of epigenetic age are indicated by a solid line while the diagonal dashed line depicts $y = x$. **d–f** Tests of the ability of the plains zebra blood clock to predict chronological age in other equids: **d** domestic horse $n = 188$, **e** Grevy's zebra $n = 5$, **f** Somali wild ass $n = 7$. **g–i** Epigenetic pacemaker (EPM) models for plains zebras. Epigenetic states (or epigenetic age) of plains zebras predicted from the EPM using **g** blood ($n = 76$), **h** remote biopsy tissue ($n = 20$), and **i** both sample types combined. Predictions are based on 76 blood samples and 20 biopsy samples. The equation of the fitted curve (solid line) is described for each plot. MAE are based on ages translated by the equation.

Supplementary Data 1. Epigenetic state was strongly correlated with chronological age in both tissue types: blood ($r = 0.97$, Fig. 1g) and biopsy ($r = 0.95$, Fig. 1h). EPMs based on the two tissues used largely distinct sets of CpG sites, sharing only 40 sites. Despite retaining a strong overall correlation in the combined EPM ($r = 0.96$), the difference in the model's performance for the two sample types is apparent (Fig. 1i). The combined tissue pacemaker shared 138 sites with blood and only 34 with biopsies.

**Association of inbreeding with biological aging.** Inbreeding was estimated both as the inbreeding coefficient F and by the proportion of the genome in runs of homozygosity (FROH). We derived the genotypes used to estimate inbreeding from two sources: RAD sequencing data (42 samples) and genotypes imputed at the same set of loci from low-coverage whole-genome sequencing data using GLIMPSE[38] (28 additional samples). GLIMPSE produced high-quality imputations (mean dosage $r^2$ of 80%; mean concordance between imputed and true genotype of

92%) as assessed by running leave-one-out imputations on 35 samples present in both the RADseq and low coverage data sets (Supplementary Fig. 2a, b). The Mendelian error rate across loci averaged 0.06 ($+/-0.06$) in the full data set. ROH were discoverable across 90% of the genome and ranged in size from 1.5 MB (minimum allowed size) up to >60 MB (Supplementary Fig. 2c). ROH over 10 MB are expected to reflect inbreeding loops occurring within the last five generations[39,40]. Individuals from more recent generations showed a pattern of excessive total ROH relative to the number of ROH segments as indicated by an upward shift in ROH size relative to number (Supplementary Fig. 2d)[39]. $F_{ROH}$ ranged from 0 to 0.37, and $F$ statistics ranged from $-0.21$ to 0.32 (one outlier with an extreme negative $F$ value was removed from the analysis). The Pearson correlation between $F$ and $F_{ROH}$ was 0.84.

We used multiple linear regressions to assess whether inbreeding is associated with age acceleration in the plains zebra population. Age acceleration was calculated by subtracting each individual's chronological age from its predicted age, and was calculated separately for the EPM and EC. Sex was added to the linear models as a covariate. EPM age acceleration was significantly associated with an interaction between chronological age and both $F$ and $F_{ROH}$ (Fig. 2 and Supplementary Table 1a, c), indicating that the impacts of inbreeding on biological age increase with chronological age. EC age acceleration was not associated with either measure of inbreeding. Sex was not associated with age acceleration in any model. A re-run of our analyses using only the RADseq samples gave similar results (Supplementary Table 1e–h).

**EWAS and functional analysis of plains zebra tissues**. We performed the EWAS analysis on the 31,836 probes that could be uniquely aligned to specific adjacent loci in the horse genome. Since the mammalian methylation array is based on stretches of DNA that are conserved in all mammals, the horse annotation can be applied to the zebra data[41]. At a nominal $p < 10^{-4}$, a total of 9757 and 331 probes were related to age in blood ($n = 76$, age-range 0.15–20.2 years) and biopsies ($n = 20$, age-range 0.16–24.8 years) respectively (Fig. 3a). The top age-related changes per tissue are as follows (Fig. 3a): blood, hypomethylation in *FANCL* upstream, *MAF* downstream, *ZNF608* upstream, and *PBX3* intron; biopsy, hypermethylation in *PLCB1*, *NEUROD1*, and *BARHL2* upstream. DNAm aging was distributed in both genic and intergenic regions relative to transcriptional start sites (Fig. 3b). Promoters and 5′UTR regions, which can be considered expression regulatory regions, mainly gained methylation with aging in both tissues. This observation paralleled a systematic positive correlation of CpG islands with age (Fig. 3c).

The association of CpG sites with chronological age in blood and biopsy was relatively similar, with a moderate positive correlation between the z-scores from the EWAS for each tissue type ($r = 0.25$, Fig. 3e). Of the CpGs with significant association with chronological age, only 81 overlapped between blood and biopsy samples (Fig. 3d). Some of these shared CpGs include hypermethylation in *PLCB1* exon, *RIMS1* exon, and hypomethylation in *NOVA2* intron and *NFIA* intron (Supplementary Fig. 3a, b). In contrast to results based on specific genes, functional enrichment analysis using GREAT[42] identified that age-related CpGs in both blood and biopsies were significantly related with regard to biological pathways, specifically development (e.g., nervous system) and survival, and were enriched with polycomb repressor complex 2 (e.g., *EED, SUZ12, PCR2*) target genes (Supplementary Fig. 4d).

## Discussion

To the best of our knowledge, this is the first study to present DNA methylation-based age estimators for any wild equid. The high accuracy of the EC models reflects that we used a custom array that profiled 36 thousand probes that were highly conserved across numerous mammalian species. These robust data allowed us to construct highly accurate epigenetic aging models for plains zebras. The best model to predict chronological age was the EC developed from blood samples, which predicted individual age with a 6-month error.

Developing a highly accurate biopsy-based EC may be challenging due to the variability of tissue types within such a sample. Biopsy samples consist of three skin layers—the epidermis, the dermis, and the hypodermis—and may even contain deeper tissues. These individual layers can exhibit different methylation patterns[43] and are often present in different proportions across samples because they vary in thickness across the body and among individuals. A comparison between blood and skin-based (dermis and epidermis only) odontocete clocks also found the skin-based clocks to be less accurate[44]. Despite the inherent difficulties of using biopsy samples and the small sample size and skewed age

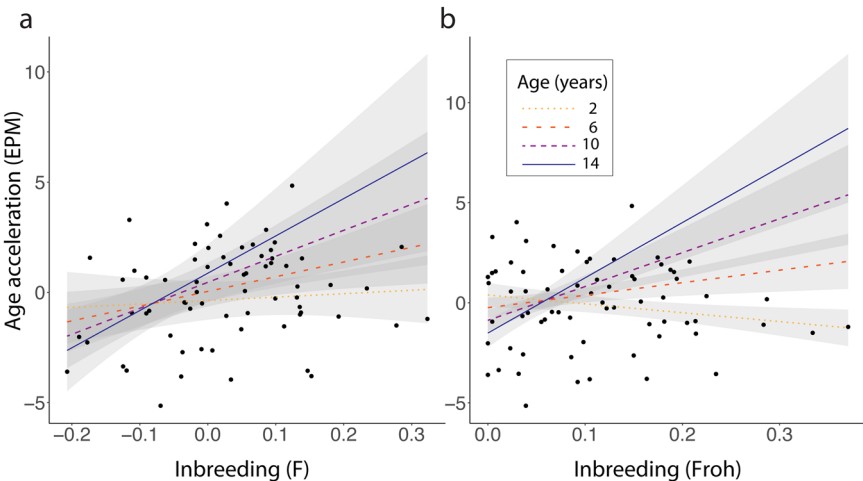

**Fig. 2 Relationship between epigenetic age acceleration calculated from the epigenetic pacemaker (EPM) model and inbreeding in plains zebras.** Lines represent the predicted age acceleration for individuals with different chronological ages and different levels of inbreeding. Gray areas show 95% confidence intervals. Black dots represent the individual plains zebra data. Inbreeding was calculated in PLINK as **a** $F$ and **b** $F_{ROH}$.

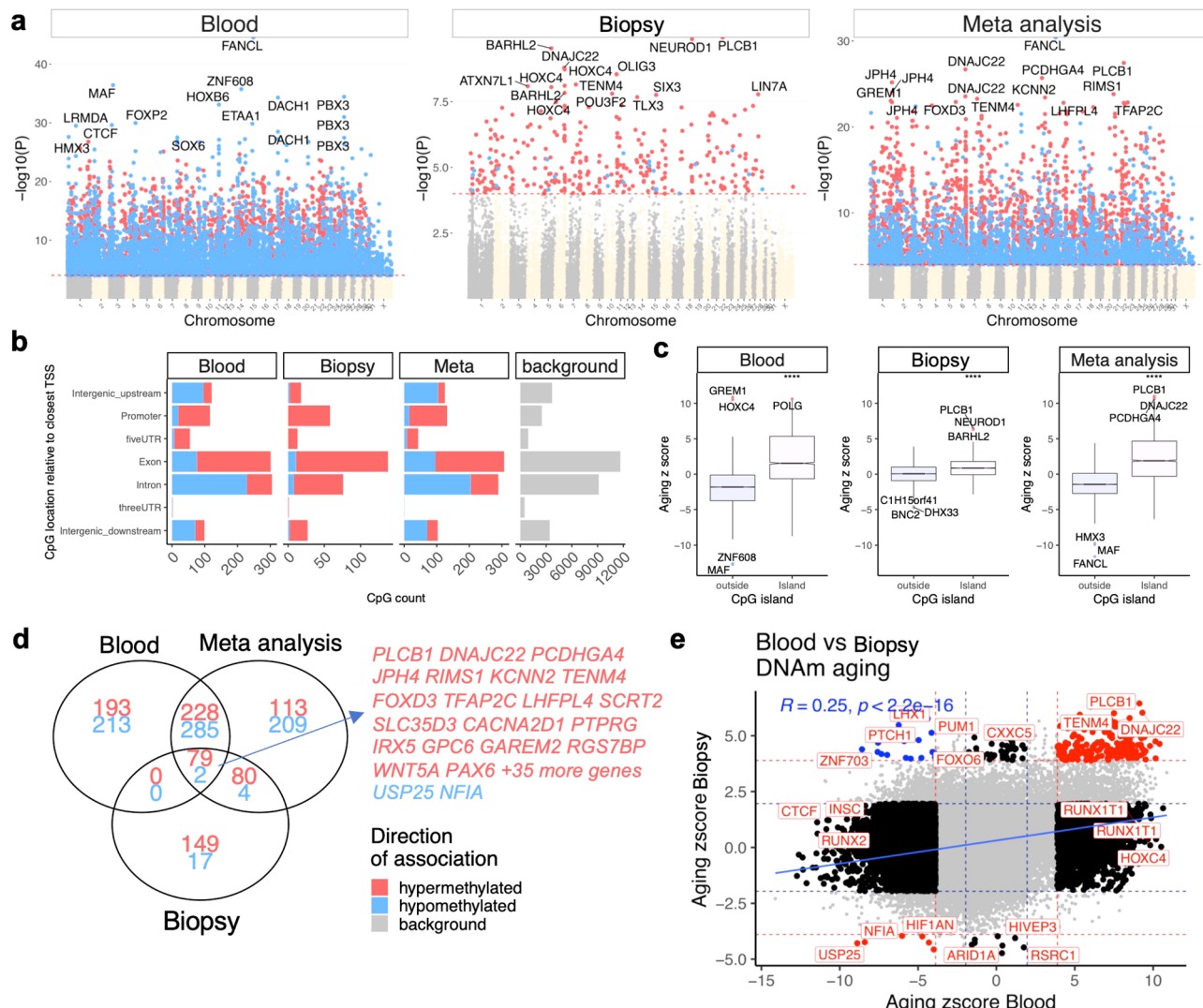

**Fig. 3 Epigenome wide association study (EWAS) of chronological age in blood and skin of plains zebras. a** Manhattan plots of the EWAS of chronological age. Since a genome assembly was not available for zebra, the coordinates are estimated based on the alignment of Mammalian array probes to EquCab3.0.100 (domestic horse) genome. The direction of associations with $p < 10^{-4}$ (red dotted line) is highlighted by red (hypermethylated) and blue (hypomethylated) colors. The top 15 CpGs were labeled by the neighboring genes. **b** Location of top age-related CpGs in each tissue relative to the closest transcriptional start site. Top CpGs were selected at $p < 10^{-4}$ and further filtering based on z-score of association with chronological age for up to 500 in a positive and negative direction. The number of selected CpGs: blood, 1000; biopsy, 331; meta-analysis, 1000. The gray color represents the location of 3,1836 mammalian BeadChip array probes mapped to EquCab3.0.100 genome. **c** Boxplot of z-scores from a correlation of age with CpG location (within or outside CpG islands). The median Z statistics are significantly different ($p < 10^{-4}$). **d** Venn diagram of the top age-related CpGs in blood and biopsy samples from plains zebras. **e** Sector plot of DNA methylation aging in plains zebra blood and biopsy tissues. Red dotted line: $p < 10^{-4}$; blue dotted line: $p > 0.05$; Red dots: shared CpGs; black dots: tissue-specific changes; blue dots: CpGs whose age correlation differs between blood and biopsy tissue.

distribution of the biopsy samples (Supplementary Fig. 5), our biopsy clock predicted age with an error of +/−2 years.

The plains zebra blood EC accurately predicted the chronological ages of horses, Grevy's zebras, and Somali wild asses. This was expected since caballine and non-caballine equids are somewhat closely related (4–4.5 MYA)[45,46]. Non-caballine species of equids are more closely related to plains zebras (1.28–1.75 MYA) than domestic horses, which may explain the lower errors found for age estimation in Grevy's zebras and Somali asses. In fact, chimpanzees and bonobos have a similar divergence time to those observed within the non-caballine equids[47] and align more closely to each other in DNAm age than either does with humans[14].

The extraordinary accuracy of ECs stems from their utilization of sites that maximize a linear relationship between epigenetic age and chronological age. This accuracy can sometimes negatively affect their ability to detect age acceleration associated with biological variation[32,33]. Methylation levels change in a non-linear fashion throughout individuals' lifetimes in several species, with accelerated changes in early life and slower changes once individuals reach adulthood[14,34]. The EPM was developed to model these non-linear changes[34,48,49]. The EPM estimates epigenetic age by maximizing the similarity between estimated and observed methylation levels, and therefore does not make any assumptions about linearity but rather identifies the shape of the relationship between age and methylation directly from the data. In this sense, the EPM is potentially more associated with biological than chronological aging[34], which may be particularly useful for investigating how environmental and life-history factors influence aging[35].

The EPM models developed here reveal that epigenetic changes occur in a non-linear fashion throughout the lifespan of plains zebras. As has been found in humans and other species[34,35], young zebras undergo faster epigenetic changes than adult zebras. We also observed that the variance in the estimates of epigenetic age is lower in young compared to old zebras. Increased variation in epigenetic age in adults is observed in humans and other species as well and may be a consequence of lifetime accumulation of environmental and physiological factors on the epigenome[7,14]. In agreement with the idea of cumulative effects on aging, the effects of inbreeding on epigenetic aging were more apparent in older individuals. We found a significant effect of the interaction between inbreeding measures and chronological age on age acceleration estimated from the blood EPM, wherein inbred individuals exhibited higher age acceleration at older chronological ages. The association between inbreeding and DNA methylation has been described in plants[28,30], chickens[31], and salmon[29]. Increases in inbreeding effects with age, such as those we describe, are predicted by theory[50] and have been shown empirically in an insect[51] and mammal[52] species.

Because the probes in the mammalian array were selected based on conservation in mammalian genomes, we expect our findings will have high translatability into humans and other mammals. The age-related gain of methylation in promoters (Fig. 3b) is consistent with observations in humans and many other species[53,54]. The low overlap of significant CpGs between tissues (Fig. 3d) may reflect the relatively low sample size ($n = 20$ skin biopsy samples) or biological differences between the biopsy and whole blood samples. Some of the genes close to the most significant CpGs in the EWAS of blood and biopsy samples play key roles in the DNA damage pathway and maintenance of genomic integrity (FANCL[55]), regulation of cellular and/or developmental processes (MAF[56], PBX3[57], NEUROD1[58], BARHL2[59,60], NOVA2[61,62], NFIA[63,64]), and extra- and/or intra- cellular signaling (PLCB1[65,66], RIMS1[67,68]).

Most of these genes are near to the top significant CpG sites from an EWAS that included multiple mammalian species and tissues: MAF, NEUROD1, BARHL2, NFIA, PLCB1, and RIMS1[69]. In mice, MAF promotes osteoblast differentiation and may be an important gene for age-related bone disease therapy[70,71]. PBX3 is at the center of the most enriched network associated with aging based on differentially methylated regions in human blood[72], and its expression levels in rat frontal cortex with aging may depend on the amount of stress experienced by the previous generations[73]. Many of these genes are potential targets for therapies to prevent and treat age-related cognitive decline and neurodegeneration. BARHL2 is differentially expressed in the hippocampus of young and old rats[74], and methylation levels in nearby CpGs are associated with aging across different tissues in naked mole rats[75]. PLCB1's expression levels are associated with aging in the human prefrontal cortex[76]. SNPs near ZNF608 have been associated with early stage of cognitive decline[77], Alzheimer's disease risk[78], and body mass index[79,80] in humans. NEUROD1 is differentially expressed with aging in the mouse hippocampus and is a critical regulator of neurogenesis[81]. RIMS1 is important for synaptic transmission at neuromuscular junctions in mammals[82]. The abundance of the NOVA2 protein in the cytoplasm decreases with aging in human neurons[83]. NFIA modulates the plasticity of local circuits in the adult hippocampus and may be involved in the cortical atrophy associated with Alzheimer's disease[84] In addition to the EWAS results, the GREAT enrichment analysis implicated developmental pathways, bivalent chromatin, and regions suppressed by polycomb repressor complex 2. These results are consistent with observations in many other mammalian species and corroborate findings on DNAm aging in many other mammalian species[3,5,14,17,32,35,69,75,85].

We expect that epigenetic aging models will be valuable for ecological studies and population management in wild species since age estimates can be used to assess reproductive potential and population viability. In known age populations, both ECs[27,41,86,87] and EPMs[34,48,49] have the potential to identify causes of individual accelerated aging. These models can be especially useful when combined with ecological or other genetic data[4]. Given the small number of CpG sites required, an aging project in a wild population could be done relatively inexpensively using bisulfite sequencing or pyrosequencing[88–90], and the costs of the mammalian array are decreasing. While epigenetic models based on blood are more accurate and less invasive than many other options for aging mammals, there are drawbacks in that animals must be immobilized to obtain the samples. Because biopsy samples can be obtained with minimal disruption[91], a highly reliable clock based on biopsy samples will be a worthwhile direction for future research. Since the methods for extracting genomic DNA from feces have improved[92–94], it will be worthwhile to explore whether epigenetic aging models can be adapted to this non-invasive source of DNA.

## Methods

**Samples**. We obtained both whole blood (96) and remote biopsy (24) samples from a captive population of zebras maintained in a semi-wild state by the Quagga Project[95] in the Western Cape of South Africa. The population was founded in 1989 with 19 wild individuals (9 from Etosha National Park in Namibia and 10 from the Kwazulu-Natal in South Africa). Since its inception, the population has undergone artificial selection to reproduce the phenotype of the extinct quagga subspecies: no stripes on legs and hindquarters, and thinner and paler stripes in the head and barrel region. At sampling, we identified individuals by their unique stripe patterns and derived their chronological ages from studbook information, in which dates of birth are typically accurate to within one month. One exception is a biopsy sample from a founder that was captured for the project as a young mare and would have been at least 25 years old at sampling. We obtained remote biopsies using an air-powered rifle affixed with a 1 mm wide by 20–25 mm deep biopsy dart and preserved in RNAlater (Qiagen). Veterinarians collected blood opportunistically during activities of the Quagga Project and preserved them in EDTA tubes. All but four samples were collected from different individuals; two individuals were sampled twice at different ages (one and three years apart, respectively). We stored all samples at −20 °C. After eliminating samples (24 of 120) with <90% confidence for individual identity or age, we retained 76 blood samples and 20 biopsy samples, totaling 96 plains zebra samples (Table 1).

The collection of 188 whole-blood samples from domestic horses is described in detail in[85]. The Grevy's zebra ($n = 5$) and Somali wild ass ($n = 7$) are samples from zoo-based animals that were opportunistically collected and banked during routine health exams. The DNA methylation profiles from these samples have been reported previously[69].

**Ethics approval**. We collected plains zebra samples under a protocol approved by the Research Safety and Animal Welfare Administration, University of California Los Angeles: ARC # 2009-090-31, approved initially in 2009.

**DNA methylation data**. We generated all DNA methylation data (plains zebra, horse, Somali wild ass, Grevy's zebra) using a custom Illumina methylation array (HorvathMammalMethylChip40)[96]. The array contains 36 thousand probes, 31,836 of which mapped uniquely to the horse genome[97,98]. We normalized methylation values from each species (plains zebra, horse, Somali wild ass, and Grevy's zebra) and tissue (blood and biopsy) using SeSAMe[99]. Unsupervised hierarchical clustering revealed that the plains zebra samples clustered by tissue (Supplementary Fig. 6).

**Epigenetic aging models**. We studied epigenetic aging in plains zebras using both EC[14,100,101] and EPM models[34,49]. For the ECs we fit generalized linear models in glmnet v.4.0-2 in R v.4.1.0[102,103]. We treated the methylation data from other equids (domestic horse, Grevy's zebra, Somali wild ass) independently and did not use them for zebra clock development. In addition to the LOO cross-validation presented here, we also conducted analyses using 10-fold cross-validation. The two forms of cross-validation did not produce appreciably different results.

We tested the ability of the blood-based EC to predict chronological age for other equids (domestic horse, Grevy's zebra, Somali wild ass) by inputting the DNA methylation profiles of these species into the plains zebra model. We used the

median absolute error of chronological age estimation and the Pearson correlation between predicted ages and known ages to assess accuracy for each species.

To construct plains zebra EPMs we used sites in which methylation levels were highly correlated with individual chronological age. The Pearson correlation ($r$) thresholds for entry into the model were absolute values of 0.75 for blood and biopsy, and 0.6 for the combined EPM. The threshold used to select the sites for input into the combined EPM was lower because only one CpG site had $r$ higher than 0.75. Epigenetic states were estimated using a leave-one-out cross-validation with EpigeneticPacemaker 0.0.3[48] in Python 3.7.4[104]. Supplementary Data 2 provides all Pearson coefficients for methylation levels against chronological age.

**Association of inbreeding with biological aging**. We prepared libraries and conducted 2 × 150 bp paired-end RAD sequencing as described in[105]. To maximize the number of SNPs ascertained, we sequenced reads produced by two restriction enzymes, SbFI and PacI (74155 and 127429 cut sites in the horse genome, respectively). We aligned reads to the horse genome EquCab3[97] using BWA[106]. We called genotypes using the haplotype-based callers Freebayes[107] and Sentieon[108], retaining only the intersection of variants called by these callers. We retained only variants within the first read of each paired-end read (a total of ~60.5 Mb). We also removed indels, multiallelic and non-autosomal sites, loci genotyped in <10% of individuals, and individuals genotyped at <20% of loci. We followed GATK's basic guidelines for additional filtering (https://gatk.broadinstitute.org/hc/en-us/articles/360035890471-Hard-filtering-germline-short-variants), excluding SNPs with QD < 2, FS > 60, SOR > 3, MQ < 40, MQRankSum <−12.5, and ReadPosRankSum <−8. Finally, we removed SNPs with MAF < 0.01. Our filtering strategy resulted in 56 individuals genotyped at 322542 loci. The number of loci is consistent with levels of heterozygosity observed in inbred wild populations of plains zebras[105].

The 56 samples with RADseq genotypes were used as a reference panel to impute these same 322542 SNPs in 89 individuals sequenced at low coverage. Low coverage libraries were constructed from 200 ng genomic DNA using Truseq Nano kit (Illumina), indexed with unique dual indices (Integrated DNA Technologies), and sequenced 24 libraries per lane on a HiSeqX platform (Illumina), generating 6.2 ± 1.4 raw Gbp per sample. We aligned sequences to the horse genome EquCab3[97] using BWA[106]. Genotypes were then imputed using GLIMPSE[38]. As part of the GLIMPSE pipeline, genotype likelihoods were called from the low coverage bams using mpileup in bcftools[109], ignoring indels and duplicate reads, and recalculating base alignment quality on the fly. Thirty-five of the 89 imputed individuals were also present in the RADseq data and were used to assess imputation quality using a leave-one-out-approach.

Forty-two individuals from the RAD-seq data and 28 from the imputed data had associated epigenetic data. The combined sample of 70 individuals spans seven generations of the Quagga Project. With the exception of removing singletons and private doubletons, we did not further MAF prune or LD prune the combined RAD and imputed data, as such pruning can bias the detection of ROH in an inbred population[110]. We used 313,645 autosomal SNPs to estimate the inbreeding coefficient F and to detect runs of homozygosity (ROH). F was estimated in PLINK[111] using methods of moments. We used PLINK's default settings to detect runs of homozygosity (ROH) with the exception that we increased the stringency to require detection of runs in 150 rather than 100 bp windows, and final runs had to be at least 1.5 MB long. In addition, we allowed only two missing SNPs per homozygous window. We converted ROH to the inbreeding coefficient $F_{ROH}$[112] by dividing the total length of ROH for each individual by the length of the genome over which we screened for ROH[110].

Linear regression models to assess the relationship between inbreeding and age acceleration were fitted with the lmtest v.0.9–38[113] package in R v.4.1.0[103]. We fitted four linear models in total, running separate analyses for the two estimates of inbreeding, F and $F_{ROH}$, and separate analyses for the two measures of age acceleration calculated based on EPM or EC. Age acceleration was the dependent variable and was calculated as the residuals of chronological age regressed on predicted age. In each analysis the independent variables were sex, chronological age, inbreeding, and the interaction between chronological age and inbreeding. We checked the residuals for normality and adjusted for heteroskedasticity via Huber-White with the package sandwich v.3.0-1[114,115]. We repeated our analyses using only the individuals genotyped directly with RAD-seq data to ensure imputed genotypes did not bias our results.

**EWAS and functional analysis of plains zebra tissues**. To identify genes potentially associated with aging, we performed EWAS in each tissue separately using the R function "standardScreeningNumericTrait" from the "WGCNA" R package[116]. The results were combined across tissues using Stouffer's meta-analysis method[117]. We estimated the distance of each significant CpG site to the closest transcription start site. We retained only the 500 CpGs with the most positive z-scores and the 500 with the most negative z-scores from each EWAS (blood, biopsy, and the combined meta-analysis). Restricting the number of analyzed CpGs did not have a drastic impact on the enriched pathway results. We used these CpGs as the input for GREAT analysis software[42]. The background was the human Hg19 genome, limited to 31,836 CpG sites mapping to the horse genome. The options in the analysis included "Basal plus extension" and a maximum of 50 kb flanking window for the CpGs coordinates.

**Statistics and reproducibility**. Statistical analyses were performed for the epigenetic models, inbreeding and age-acceleration, and EWAS and functional analysis. The analyses are described in the corresponding Methods sections, including all parameters used to allow reproducibility.

**Reporting summary**. Further information on research design is available in the Nature Research Reporting Summary linked to this article.

## Data availability
Methylation data for plains zebras can be downloaded from Gene Expression Omnibus GSE184223. RAD sequencing data is available as fastq files on SRA, BioProject ID: PRJNA670933. Data and scripts associated with model development and the creation of Figs. 1 and 2 are published on DRYAD (doi:10.5068/D1W39K).

## Code availability
Scripts used to run EWAS and functional analyses are at https://github.com/shorvath/MammalianMethylationConsortium.

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

## Acknowledgements

This work was supported by the Paul G. Allen Frontiers Group (SH). G.P. was supported by the Science Without Borders program of the National Counsel of Technological and Scientific Development of Brazil. Sample collection was supported by National Geographic Grant 8941-11 (BL). We thank the following people for their assistance with acquiring the plains zebra blood and tissue samples: Colleen O'Ryan, Stephen Mitchell, Mick D'Alton, Tom Turner, Boet le Roux, Evert Grobbelar, Ewald Groenewald, Basie and Coenraad Bezuidenhout, Linda Mason, Patricia Swanepoel, Fernando Rueda, Ross Cowlin, Melissa Stander, Hanna Lindstadt, Ansel Abels, J.P. Hugo, Cobus van Coller, Jannie du Plessis, and SANParks. White Oak Conservation kindly provided Grevy's zebra and Somali wild ass samples.

## Author contributions

B.L., S.H. and G.P. conceived of the study. B.L. and G.P. analyzed data, and B.L., G.P., and S.H. co-wrote the article. The remaining authors helped with the statistical analysis (A.H., J.Z., C.L., C.F., and M.P.), or the data generation (B.W., C.J.F., C.K., G.B., T.R., and D.M.). All authors reviewed and edited the article.

## Competing interests

The authors declare the following competing interests: S.H. is a founder of the non-profit Epigenetic Clock Development Foundation which plans to license several patents from his employer UC Regents. These patents list S.H. as inventor. The other authors declare no competing interests.
