## [Peer Review File · Communications Biology]

Reviewers' comments:

Reviewer #1:

In this study, the authors developed epigenetic clocks for plain zebras and tested the accuracy of these models in other equid species. The study is set up in a clear and straightforward manner and the results are thoroughly described. The manuscript is a clear, exciting step towards possible usage of epigenetic clocks in wildlife management.

I have a few minor comments:

1. Please justify why the 500 CpGs with highest positive z-scores and 500 with negative z-scores were used for the GREAT analysis.

2. Please discuss the genes with age-associated DNA methylation changes following EWAS analysis. Did any of the identified genes/regions overlap with age-related genes in other species?

Reviewer #2:

Overall the paper by Larison et al. is well written and uses a cutting-edge assay and model. The approach is supported and hypotheses justified; I do have some concerns about the genomic data and importantly the FROH estimates that I think should be addressed before we confidently make a link to accelerated DNAm age and inbreeding - which will be very exciting and important.

L200 - do you mean the residuals?

Genomic data - overall I found this section the most underwhelming and potentially problematic. Freebayes on RADseq is uncommon and generating 1 million variants is almost an order of magnitude higher than what we see in most mammals (especially those with inbreeding coefficients presented here). Freebayes is very liberal and I'd like to see more standard approaches like STACKS or even ANGSD applied. Why use genotype likelihoods with mpileup for WGS but not for RADseq? There's plenty of studies out there showing a caller effect. I would encourage the authors to match up the callers, show the F metrics aren't impacted by caller choice.

How many of these SNPs are seen in 1 individual? No MAF filter was applied, but I worry the majority of these SNPs are in 1 sample and possibly due to the use of Freebayes. Line 43 of supp. what is G when referring to read number? What does "generally high quality imputations" actually mean? Some individuals had hets imputed at ~50% success- I would think this is a problem for ROH estimates. Moreover, compounded across the data set I would think that 4 of 7 individuals with low het imputation (my guess from the figure) and 20% error overall error, leads to a less than ideal data set for FROH. What was the correlation between F and FROH - this would be useful to know and appease my concerns over your ROH estimates.

Last main point here - there was an association between an interaction; FxAge, but the original regression has age, so you're modeling the difference between true age and DNAm age - it's not clear to me why age is then included in this model - or at least biologically what does this mean? The effect of inbreeding on DNAm increases with age? This just needs to be flushed out for the reader to understand.

Minor.

Froh: numerator should be length of scaffolds screened - this may or may not be the entire genome. please clarify.

L212; how many sites on average did you need to impute? what was the coverage of these WGS samples? I'd like to know how many sites were imputed as ideally this was a small fraction.

L214: are these 300K SNPs from the RADdata? That's a lot for RADseq, especially if you think inbreeding is at play. Most vertebrates with standard RADseq protocols produce 10,000s

Below are our responses (in blue) to reviewer comments. The line numbers of the manuscript where changes can be seen refer to the clean document.

Reviewer #1:

In this study, the authors developed epigenetic clocks for plain zebras and tested the accuracy of these models in other equid species. The study is set up in a clear and straightforward manner and the results are thoroughly described. The manuscript is a clear, exciting step towards possible usage of epigenetic clocks in wildlife management.

I have a few minor comments:

1. Please justify why the 500 CpGs with highest positive z-scores and 500 with negative z-scores were used for the GREAT analysis.

Response: We limited the input to 500 CpGs because we were interested in understanding the biology of age-related changes in the most significant CpGs. There is also a danger that a larger number would increase the noise and hence lead to false positive results in the GREAT analysis. We now state this in the manuscript on lines 259-261. We also noticed that restricting the number of analyzed CpGs had no drastic impact on the enriched pathway results. We also decided to perform the analysis separated by the direction of methylation change because it can inform us about a potential alteration in the biological processes: activation or suppression of pathways. Work in other species has revealed different biological processes. Gain of methylation occurs in CpG islands and bivalent chromatin regions that play a role in development. Age related loss of methylation often occurs outside of CpG islands, in heterochromatin, and enhancers.

2. Please discuss the genes with age-associated DNA methylation changes following EWAS analysis. Did any of the identified genes/regions overlap with age-related genes in other species?

Response: Thank you for this comment. We added additional information about age-related genes in lines 403-424.

Reviewer #2:

Overall the paper by Larison et al. is well written and uses a cutting-edge assay and model. The approach is supported and hypotheses justified; I do have some concerns about the genomic data and importantly the FROH estimates that I think should be addressed before we confidently make a link to accelerated DNAm age and inbreeding - which will be very exciting and important.

L200 - do you mean the residuals?

Yes that is what is meant by chronological age subtracted from predicted age – for clarity we have changed line 244 to read “the residuals of chronological age regressed on predicted age”

Genomic data - overall I found this section the most underwhelming and potentially problematic. Freebayes on RADseq is uncommon and generating 1 million variants is almost an order of magnitude higher than what we see in most mammals (especially those with inbreeding coefficients presented here). Freebayes is very liberal and I'd like to see more standard approaches like STACKs or even ANGSD applied. Why use genotype likelihoods with mpileup for WGS but not for RADseq? There's plenty of studies out there showing a caller effect. I would encourage the authors to match up the callers, show the F metrics aren't impacted by caller choice.

We appreciate the concerns of the reviewer about the apparent large number of variants and differences among callers. Since the details are likely to be of concern to readers, we have

enhanced our description of the process and moved it from the Supplement to the main paper (Lines 198-240). We also endeavor to clarify point by point here:

Freebayes and the expected number of variant sites: We actually used two variant callers, Freebayes and Sentieon (a commercial version of GATK), and retained only those variants that the two callers agreed upon. After filtering, which we now more thoroughly describe, we retained 313435 SNPs for subsequent analyses.

The expected number of variants can be roughly estimated from genome coverage x heterozygosity. The amount of genome coverage is 60,475,200 nt, derived from 201584 restriction sites (74155 from Pac1 and 127429 from Sbf1) and 300 nt of sequence per cleaved restriction site. Based on data from an earlier survey of 100 plains zebras including several isolated small populations (Larison et al., 2020), we observed individual heterozygosity values of 0.0017 - 0.0027. Using the more conservative value of 0.0017 suggests that we might observe 102,807 variant sites in a single individual, and a tally of variants in a sample of 10 individuals from an inbred site had 332613 variant sites. Thus, observing 322532 variants in 56 individuals (as in the current work) is well within expectations. In addition, despite inbreeding levels observed in more recent generations of the Quagga Project zebras, the observed variant count is likely to be higher than values estimated from a single population, because the project was started by intercrossing individuals from two disparate populations.

In addition, we have checked for mendelian inconsistencies in the data and found very low instances of mendelian error in both the RADseq (0.04) and imputed data (0.06) as reported on line 297.

Use of mpileup for both RAD genotyping and imputation: There are a couple of arguments we feel are important to consider here. First, there are intrinsic differences between RADseq and low coverage WGS that justify, and indeed necessitate different approaches to obtaining genotypes. Our RADseq data is high depth (~30X) such that genotypes can be accurately called. Genotypes cannot be called using low coverage data (~2X) and therefore one must resort to using genotype likelihoods. Imputation must then be used to generate genotypes in the low coverage data. Second, bcftools mpileup is a naïve caller that considers information at each base along a set of sequences and is therefore quite liberal. Freebayes and Sentieon are both haplotype-based callers that take into account sequence information around each variant and are therefore inherently more conservative. GLIMPSE uses mpileup to call genotype likelihoods from which genotypes are called based on a reference panel of trusted genotypes. Given our reference panel was the set of individuals with RADseq data it makes more sense to use the approach we did, which was rigorous and conservative, than to use mpileup.

How many of these SNPs are seen in 1 individual? No MAF filter was applied, but I worry the majority of these SNPs are in 1 sample and possibly due to the use of Freebayes.

Thank you for pointing out this lack of clarity on our part. A minimal MAF filter (0.01) was applied to the RADseq data but not once the RAD and imputed data were joined due to concerns about biasing the data in an inbred population as we state on lines 231-233. However, a search using VCFTools did find some singletons and private doubletons (210 in number) which we removed prior to rerunning the analyses. This information has been added to the manuscript at line 232.

Line 43 of supp. what is G when referring to read number?

Thank you for catching that typo. This has been rewritten to say "6.2±1.4 raw Gbp per sample"

What does "generally high quality imputations" actually mean? Some individuals had hets imputed at ~50% success- I would think this is a problem for ROH estimates. Moreover, compounded across

the data set I would think that 4 of 7 individuals with low het imputation (my guess from the figure) and 20% error overall error, leads to a less than ideal data set for FROH.

We agree this statement is rather vague and have added some details on lines 294-297. The mean dosage r^2 was 80%, but the mean concordance between the imputed and true genotype was 92%. Overall, four individuals out of the 35 test individuals imputed poorly (11%) and that is a concern. To address the concern over imputation and our estimates of ROH being biased we re-ran the F and Froh estimates using only RADseq data. We used these new estimates to rerun regressions. The results using RADseq tell the same story overall with some difference in significance levels. This is discussed on lines 250-252 and 310-311, and the RADseq results have been added to the table in the supplement.

What was the correlation between F and FROH - this would be useful to know and appease my concerns over your ROH estimates.

The correlation between F and FROH in the full data set (RAD + imputed) was 0.84 – this information has been added to the manuscript at line 304.

Last main point here - there was an association between an interaction; FxAge, but the original regression has age, so you're modeling the difference between true age and DNAm age - it's not clear to me why age is then included in this model - or at least biologically what does this mean? The effect of inbreeding on DNAm increases with age? This just needs to be flushed out for the reader to understand.

The reviewer has understood this correctly and we clarify it on lines 307-308 by directly stating that our findings 'indicate that the impacts of inbreeding on DNA methylation increase with age'

Minor.

Froh: numerator should be length of scaffolds screened - this may or may not be the entire genome. please clarify.

It was the length of scaffolds screened as outlined in reference 58. Lines 238-240 now read: "We converted ROH to the inbreeding coefficient F_{ROH}^{60} by dividing the total length of ROH for each individual by the length of the genome over which we screened for ROH⁵⁸."

L212; how many sites on average did you need to impute? what was the coverage of these WGS samples? I'd like to know how many sites were imputed as ideally this was a small fraction.

The coverage of the WGS samples was ~2X. The imputation approach used here was not aimed at imputing a few poorly typed SNPs in a few individuals. The imputation approach we used is designed to impute a large number of SNPs in previously ungenotyped individuals using a reference panel. We have endeavored to clarify this in the expanded methods section. In particular lines 199-201, 218-219, and 226-230 should help clarify that we imputed all genotypes for entire samples

L214: are these 300K SNPs from the RADdata? That's a lot for RADseq, especially if you think inbreeding is at play. Most vertebrates with standard RADseq protocols produce 10,000s

Please see our response to the second question – about Genomic Data. This is covered there.

References

Larison, B. *et al.* Population structure, inbreeding and stripe pattern abnormalities in plains zebras. *Mol Ecol*, doi:10.1111/mec.15728 (2020).

REVIEWERS' COMMENTS:

Reviewer #2 (Remarks to the Author):

I am overall satisfied with the revisions and response.

REVIEWERS' COMMENTS:

Reviewer #2 (Remarks to the Author):

I am overall satisfied with the revisions and response.

Response: we thank both reviewers for their valuable help in improving our manuscript.